# High Molecular Weight AB-Polybenzimidazole and Its Solutions in a Complex Organic Solvent: Dissolution Kinetics and Rheology

**DOI:** 10.3390/polym14214648

**Published:** 2022-11-01

**Authors:** Ivan Y. Skvortsov, Lydia A. Varfolomeeva, Igor I. Ponomarev, Kirill M. Skupov, Aleksandra A. Maklakova, Valery G. Kulichikhin

**Affiliations:** 1A.V. Topchiev Institute of Petrochemical Synthesis of Russian Academy of Sciences, Leninsky Av. 29, 119991 Moscow, Russia; 2A.N. Nesmeyanov Institute of Organoelement Compounds of Russian Academy of Sciences, Vavilova St. 28, Bld. 1, 119334 Moscow, Russia

**Keywords:** Poly(2,5(6)-benzimidazole, ABPBI, solubility, complex superbasic medium solvents, rheology, polymer dopes for spinning

## Abstract

AB-polybenzimidazole (ABPBI) dissolution kinetics in an eco-friendly complex acid-free solvent based on dimethyl sulfoxide (DMSO), methanol and KOH, and the rheological behavior of their solutions are investigated. The optimal component ratio of solvent providing the complete ABPBI dissolution is determined. Methanol containing dissolved KOH contributes to the creation of a single-phase superbasic medium, which accelerates and improves the polymer solubility in a mixture with DMSO, significantly reducing the viscoelasticity of the resulting solution. The optimum methanol content is up to 60 wt.% related to DMSO. The polymer dissolution rate increases by 5 times in this composition. It found the polymer concentration of 9% is close to the dissolution limit due to the strong solution structuring, which is probably associated with an increase in the amount of water released during the KOH-methanol-DMSO interactions. As a result, the conditions for obtaining high concentrated solutions in a complex, mainly organic solvent for fiber spinning are developed. The viscoelastic properties of solutions are measured in the concentration range of 1–9% at temperatures of 20–50 °C. The flow activation energy for 7 and 9% solutions decreases by 1.5 and 2.3 times, respectively, as the content of methanol in the complex solvent increases from 10 to 60%.

## 1. Introduction

ABPBI is a high-molecular semi-rigid (Kuhn segment 7.9 nm) [1] polymer belonging to the polybenzimidazole (PBI) family. It is one of the most promising classes of polyheteroarylenes, which are of great interest due to a set of unique properties: thermal, heat-, fire [2,3,4], and chemical resistance, the highest equilibrium moisture content among polyheteroarylenes (about 15%), and potentially low cost [5]. Usually, only natural fibers like cotton possess such water uptake properties, which makes them preferable to produce special comfortable clothes with a high fire resistance. The high equilibrium moisture content is due to the presence of a mobile proton near benzimidazole nitrogen atoms, with a lone pair of electrons in the ABPBI molecule [6,7].

The PBI family is characterized by high glass transition temperatures, often exceeding 450 °C [8,9], good mechanical properties, and excellent chemical stability which, on the whole, is the main reason to use them as materials for extreme conditions [9,10]. PBI polymer has a wide range of applications: for the manufacture of capacitive deionization anion exchange membranes [11,12], for the production of high chemical resistance in various organic solvents membranes for nanofiltration with good separation performance [13], and as a component for the production of proton-conducting membranes [14]. The most common and commercially available PBI (Celazole^®^) is synthesized from the expensive and carcinogenic 3,3′-diaminobenzidine and isophthalic acid diphenyl ester [2,4].

For the ABPBI synthesis, cheap and commercially available 2,4-Diaminobutyric acid (DABA) is used [1,2,8,15,16]. During the synthesis, there is no need to observe stoichiometric ratios, which is also advantageous compared with traditional technologies for PBI production [1,2,15,17]. An effective and simple method for purifying DABA by converting it into DABA monophosphate (MP) was developed in [2], which allows to improve the monomer solubility, avoid oxidative degradation processes, and obtain ABPBI with high molecular weights (up to 120 kg∙mol^−1^) [1]. Thus, ABPBI solutions can be successfully used to obtain products such as fibers or films with a sufficiently high level of strength [15].

Until now, the production of fibers and films from a high molecular weight ABPBI was carried out from acid solutions with concentrations lower than 5.5% [6,7]. The processing was limited by the fact that the ABPBI spinning solution dopes were based on such aggressive solvents as methanesulfonic acid [6,18], polyphosphoric acid (PPA), or a mixture of P_2_O_5_ with methanesulfonic acid (Eaton’s reagent) [11,19]. A low molecular weight ABPBI (*M_ꞷ_* ≤ 50−60 kg∙mol^−1^) can be dissolved in sulfuric acid [8], trifluoroacetic acid [19], methanesulfonic acid [8], formic acid [8], N-methyl-2-pyrrolidone (NMP)/LiCl [20] and ethanol/NaOH mixtures [20].

The coagulation process of the ABPBI solutions in methanesulfonic acid with deionized water at film preparation was described in [19]. The polymer concentration in the solution was 2.5%, which leads to high film shrinkage. At the same time, no through porosity was found in the films, which indicates the suitability of water and aqueous solutions of phosphoric acid as coagulants.

The data on the solubility of the rather low molecular weight ABPBI (*M_ꞷ_* ≤ 50−60 kg∙mol^−1^, intrinsic viscosities 1.4−2.0 dL∙g^−1^) in NMP, and properties of solutions for film preparation have been published in [21]. The authors obtained solutions with different polymer concentrations (4–13%), studied their rheological behavior, and evaluated the effect of solution viscosity on the morphology of the resulting films. It was shown that an increase in the solutions’ viscosity leads to a decrease in the number of pores on the film surface, and a decrease in the size of macropores in the film cross-section from 10 µm to 2–3 µm. In the same work, the effect of temperature and air humidity on the nature and size of pores formed in ABPBI films was studied. In particular, water vapor was used as a coagulant, and its temperature and the value of the relative humidity of the environment were varied during the coagulation process to find appropriate conditions for reaching the desired film porosity.

For the synthesis of ABPBI in Eaton’s reagent [15], the ABPBI fibers and films were spun by immersing a layer of the reaction solution on a glass plate in water. However, optical images of the resulting film indicate a non-uniform morphology, which causes the fragility of the film.

A new complex organic solvent based on a DMSO (dimethyl sulfoxide): MeOH (methanol): KOH (potassium hydroxide) mixture was developed in [1,2], which marks a new approach to the dissolution of high molecular weight (*M_ꞷ_* up to 120 kg∙mol^−1^) ABPBI, that opens up wide possibilities for processing these solutions into high-strength fibers and films.

Thus, based on the brief review, to obtain high-strength fibers it is necessary to work with a high molecular weight polymer [22], and it is advisable to spin the fibers from highly concentrated solutions. Namely, this combination of polymer and solution characteristics makes it possible to exclude defects caused by both coagulation processes [23,24,25] and uneven shrinkage [19], leading to the formation of fibers with a non-circular cross-section and a corresponding deterioration in mechanical properties [26].

This work is thus devoted to a comprehensive study of the main solution parameters for high molecular weight ABPBI in a complex organic solvent, which determines the effective fiber spinning partially described in [1,2]. These include the ratio of solvent components, the dissolution kinetics in the chosen coagulant, polymer concentration, and rheological properties of solutions at different temperatures.

## 2. Materials and Methods

### 2.1. ABPBI Synthesis

A 25% solution of DABA MP in 84% PPA was used for the synthesis. The synthesis is described elsewhere [1,2]. Briefly, it was carried out in Teflon cups in a muffle furnace with stepwise heating from 120 to 180 °C in argon flow. The scheme of the synthesis is shown in Appendix A (Figure A1).

### 2.2. Preparation of ABPBI Solutions

ABPBI powder was preliminarily dried for 3 h at 150 °C under vacuum (residual pressure 0.05 bar) to a constant weight, followed by dissolution in a complex organic solvent based on a DMSO:MeOH:KOH mixture. The polymer was poured into a glass vial, solvent components were added, and purged with N_2_. Next, the vials were sealed and compositions were constantly stirred with a J-shaped stirrer at a speed of 5–20 rpm at 50 °C until the polymer was completely dissolved (16–80 h, depending on the concentration). As a result, a wide range of solutions with different ABPBI polymer contents (3–9 wt.%), as well as with different DMSO:MeOH ratios (Table 1) was prepared. The amount of KOH was 50 wt.% of the polymer in all cases.

The dissolution process slows down with an increase in the polymer concentration and a decrease in the methanol content in the solvent. Thus, for the 9% composition A9, the dissolution took place within two weeks, followed by homogenization and filtration of the solution.

For the dissolution test, ABPBI fibers obtained from an A10 solution with a diameter of 30 µm were used.

### 2.3. Rheology

Rheological studies were carried out on a HAAKE MARS 60 rheometer (Thermo Fisher Scientific, Karlsruhe, Germany) using the following geometries:-cone-plate with a diameter of 20 mm and an angle between the cone and the plate of 1° (for solutions containing 9% of ABPBI);-cone-plate with a diameter of 60 mm and an angle between the cone and the plate of 1° (for solutions containing 7% of ABPBI);-a bicone with a diameter of 60 mm and an angle between the cone and the plates of 1° (for solutions with ABPBI content of 3%).

A protective solvent hood was used to avoid the solution gelation by the air moisture during the experiment.

In the stationary deformation mode, the flow curves were obtained in the range of shear rates 10^–1^–10^3^ s^–1^. To determine the area of linear viscoelasticity for the subsequent measurement of the frequency dependences of the storage and loss moduli, the complex modulus of elasticity was preliminarily measured in the strain range of 0.1–100% at frequencies of 1 Hz (6.3 rad∙s^−1^) and 80 Hz (503 rad∙s^−1^). The frequency dependences of the storage and loss moduli in the linear region were measured in the frequency range of 0.628–628 rad∙s^−1^.

The intrinsic viscosities were measured by an Ubbelohde capillary viscometer at 25 °C following ASTM D2857 [27].

### 2.4. Polymer Dissolution Kinetics

ABPBI dissolution kinetics in DMSO:MeOH:KOH complex solvents with various DMSO:MeOH ratios equal to 9:1; 7:3 and 4:6 was examined using optical microscopy. An ABPBI monofilament fiber of equal length (40 ± 0.5 mm) and diameter (28 ± 0.4 μm) was placed between two glass slides, a solvent was added to a gap equal to the fiber thickness, and the dissolution process was observed using a Biomed 6PO microscope (Biomed Co, Moscow, Russia) with the camera ToupTek E3ISPM5000 (ToupTek Photonics Co., Hangzhou, China), taking images of the process at regular intervals for each solvent. The moment of solvent contact with the fiber surface was chosen as the starting dissolution process time. The corresponding measuring scheme is shown in Figure A2 in the Appendix A. The experiment was repeated at least 5 times with each solvent composition to obtain statistical massive data on fiber dissolution. The illustrative part presents the images of the most representative samples.

### 2.5. Determination of Water Content

Residual moisture content was determined by a coulometric titration using an Expert-007M instrument manufactured by Econiks-Expert LLC (Moscow, Russia).

### 2.6. XRD Analysis

The structure of the samples was studied by X-ray diffractometry on a Rigaku Rotaflex D/MAX-RC setup (Rigaku Corporation, Tokyo, Japan), equipped with a rotating copper anode (the operating mode of the X-ray source was 50 kV, 100 mA, the characteristic radiation wavelength was *λ* = 0.1542 nm, a horizontal goniometer, and a scintillation detector). An X-ray survey was carried out in the “reflection” and “transmission” geometry, according to the Bragg–Brentano scheme in the continuous θ–2θ scanning mode in the angular range of 3–45° and the scanning step of 0.04° at room temperature. To obtain diffractograms, parallel bundles of the fibers of their fragments (~100 pieces) were used in the vertical or horizontal directions relative to the axis of the goniometer.

## 3. Results and Discussion

### 3.1. Composition of the Multicomponent Solvent

An ABPBI elementary unit has one –N-H-group in the benzimidazole cycle, capable of active solvation (Figure A1) [6,7]. Therefore, the presence of a superbase is necessary for its dissolution and deprotonation as well. It is a strong complex (Bronsted base) with a ligand that reacts with a cation of this base (Lewis base). In an aqueous medium (as methanol), the formation of a superbase is impossible due to its low upper acidity limit (pKa < 16). At the same time, DMSO weakly solvates anions and has a high basicity limit (pKa = 35), which makes such a solvent an excellent medium for obtaining a superbase. Unfortunately, the actual values of the acidity of DMSO/KOH solutions are usually significantly lower due to the release of water during the formation of dimsyl-potassium [28].

KOH is almost insoluble in DMSO (only 0.013 g∙dL^−1^ at 25 °C) [29], which makes it impossible to use this mixture without an additional compatibilizing agent, soluble both in alkali and in aprotic solvent, since it needs at least an equimolar ratio of potassium ions to the ABPBI unit. Methanol was used as a cosolvent. A hypothetical scheme of polymer solvation is shown in Figure 1.

Methanol reacts reversibly with KOH, forming potassium methoxide with the water release. It is a very strong base. Thus, when KOH, MeOH, and DMSO are added to the system, a complex of superbasic compounds is formed: potassium methylate and dimsyl, and the basicity of the medium is mainly determined by the presence of water released during the reaction.

The water content in a mixture of pre-dried DMSO:MeOH solvents was determined by coulometric titration in a ratio of 4 to 6. In such a system, the amount of released water is 0.017 wt.%. After the addition of 1.5% KOH into this mixture, the water content increases to 0.61%, which in molar ratio corresponds to the calculated amount of water released during the reaction of methanol with KOH to form potassium methoxide.

It is known that the basicity of the aprotic medium significantly depends on the amount of water [28], decreasing its content of ~1% from 35 to 30–32 and ~26 at a water content of 5%. Following this, it can be expected that a lack of potassium methoxide/dimsyl in the system will reduce the dissolving power due to the limited solvation of the polymer, while an excess of this complex will significantly reduce the basicity of the medium up to the exit from the superbasicity region (pKa less than 20), with a corresponding termination of polymer solubility.

Thus, to obtain the best solvent for ABPBI, it is necessary to create a three-component solution, with the minimum sufficient methanol content necessary for the complete dissolution of alkali in DMSO and the formation of a single-phase three-component solvent with high basicity.

Consider in turn the principles of choosing the content of the solvent components.

#### 3.1.1. KOH

Based on the above information, it was experimentally confirmed that ABPBI does not dissolve at any ratio of MeOH and DMSO in the absence of alkali. The addition of KOH leads to the gradual dissolution of the polymer, with complete dissolution starting from the equimolar ratio of KOH to the ABPBI unit. An excess of KOH does not affect the polymer solubility but can serve as a source of defects in the as-spun fiber during the subsequent washing out of the alkali. To reduce the mass concentration, it would be preferable to use lighter elements (eg sodium or lithium hydroxides). However, they have a significantly less density charge (due to the smaller mass of the atom) and solvate the polymer less effectively.

#### 3.1.2. Methanol

To determine the optimal ratio of MeOH to DMSO in a complex solvent, at the initial stage, the boundary maximum and minimum concentrations of methanol were determined at constant concentrations of ABPBI (7%) and KOH (50 wt.% of the polymer). The morphology of the systems is shown in Figure 2.

When the methanol content is less than 6%, an insufficient superbase amount, potassium methylate/dimsyl is formed. As a result, complete solvation of the polymer does not occur and only partial dissolution is observed, as can be seen in Figure 2a. The complete dissolution of the polymer occurs in the range of DMSO:MeOH from 94:6 to 30:70 (Figure 2b,c), while the dissolution rate and rheological properties of the resulting solutions change significantly, as will be shown below. With an increase of methanol to a ratio of DMSO:MeOH 26:74 (Figure 2d), the polymer is dissolved only partially with the formation of many swollen gel-like particles (examples are shown by an arrow). The gradual deterioration of the polymer solubility at high concentrations of methanol is probably associated with a significant decrease in the basicity of the medium, and the corresponding inability of the polymer to be solvated by the formed weak base.

### 3.2. Viscometry of Solutions

The ABPBI inherent viscosity in a complex solvent (DMSO:MeOH:KOH, DMSO:MeOH ratio = 4:6, KOH 100 wt.% of the polymer) was determined using the Huggins Equation (1) [30]:(1)$$\frac{\eta_{sp}}{c}=\left[\eta \right]+k_{H}{\left[\eta \right]}^{2}\;$$
where *η_sp_* is the specific viscosity, *[η]* is the intrinsic viscosity, *k_H_* is the Huggins constant, and *c* is the concentration of the solution.

The dependence of the reduced viscosity on the concentration is shown in Appendix A (Figure A3).

The resulting linear dependence, reduced to zero concentration, allows us to determine the values of intrinsic viscosity *[η]* = 4.9 dL∙g^−1^ and Huggins constants equal to 0.5 and 0.66 for solutions in complex solvents with DMSO:MeOH ratios equal to 9:1 and 4:6, respectively. In accordance with the data obtained in our previous work [2], this intrinsic viscosity value corresponds to a *M_wD_*~100 kg·mol^−1^. An increase of the methanol fraction in the system reduces the overall basicity of the solvent and leads to deterioration in the affinity of the solvent for the polymer, which can be seen from the increase of the Huggins constant. For this polymer, the transition region between dilute and semi-dilute solutions (crossover concentration, or *c**), calculated as *c*·[η]* ≈ 1 according to [31], occurs at a concentration of ≈0.2 g∙dL^−1^.

Figure 3 shows the results of measuring the viscosity of ABPBI solutions in the concentration range from dilute to concentrated solutions as a function of the reduced viscosity on the dimensionless parameter *c[η]*, which is the volume occupied by macromolecules in the solution.

The nature and composition of the complex solvent make it possible to compare the behavior of solutions lying on a single, broken line with the characteristic slopes of its sections. The slope in the region of low concentrations is equal to 1.5. This indicates some deviation from the regime usually attributed to ideal dilute solutions, for which the slope should be equal to unity [32]. Perhaps this is due to specific interactions in the system with a complex solvent, or to the high rigidity of macromolecules, for which hypothesis on the interpenetration of coils turns out to be only a certain approximation to reality. The transition to the region of concentrated solutions is characterized by a three-dimensional network formation with physical contacts. These entanglements are either dispersion (van der Waals) or ionic interactions, which are reflected in this dependence by a sharp increase in the slope to ≈4.7, corresponding to the literature data for concentrated solutions of different polymers [31,32,33]. It can be seen that the behavior of a 9% solution in a 9:1 DMSO:MeOH mixture starts to deviate from the general relationship, indicating that the limiting solubility of the polymer in this solvent has been reached, and is probably caused by a lack of soluble free alkali in the solvent.

### 3.3. Polymer Dissolution Kinetics

One of the best direct methods for a dissolution kinetics investigation, which allows us to directly evaluate the dissolution kinetics by solvent penetration into the polymer with corresponding diffusion front propagation [34], is using an interferometry method [35]. This turned out to be inapplicable for even thin (less than 100 µm) dried ABPBI films. Therefore, the method [36] of assessing the rate of swelling and subsequent dissolution of a thin, homogeneous fiber was applied to determine the composition that almost instantly dissolves the fiber, and to estimate the solvent quality this way. This method was particularly clear when dissolving the dark and contrasting ABPBI fiber, which made it possible not only to evaluate the kinetics of fiber thinning, but also to determine the difference in the rate of the solvent diffusion front.

ABPBI fiber dissolution kinetics in a DMSO:MeOH:KOH complex solvent, with an excess of KOH related to the polymer content (i.e., from a known fiber weight) at different ratios of DMSO to MeOH (9:1, 7:3; 4:6), is shown in Figure 4.

The activity of potassium methoxide in the complex solvent increases with an increase in the fraction of methanol, which affects the polymer dissolution rate. Therefore, in a solvent with a ratio of DMSO:MeOH 9:1, the time of complete dissolution of the fiber (i.e., the moment of disappearance of the visible fiber boundary) is ~410 s (Figure 4a). For a ratio of 7:3 (Figure 4b), it decreases by almost 3.5 times and is already 120 s. The highest dissolution rate is observed for the ratio DMSO:MeOH 4:6 (Figure 4c): the complete dissolution time is equal to 75 s. Thus, the dissolution rate increases by 5.5 times with an increase in the methanol content in the complex solvent.

The obtained images were processed to determine a clear boundary of the diffusion front (Figure A4 in Appendix A). Based on these data, the corresponding profiles of the thinning of the dissolving fiber (Figure 5a) and the polymer diffusion front propagation in the solvent (Figure 5b) were obtained.

It can be seen that in all cases, the dissolution process proceeds in three stages. The initial thinning of the fiber, due to the onset of dissolution of its outer shell, is accompanied by simultaneous swelling of the bulk polymer in the solvent (stage I in Figure 5a). The rate of swelling then begins to prevail over the rate of dissolution, which is reflected in a partial increase in the fiber diameter (stage II in Figure 5a). Finally, the swollen fiber starts to dissolve rapidly (stage III in Figure 5a). The calculated rates of fiber thinning in stages I and III are shown in Table 2.

It should be noted that regardless of the solvent used, the maximum diameter observed at the beginning of stage III is observed at about the same time, while the time for a complete dissolution of the fiber differs by more than 4 times for solvents 9:1 and 6:4, respectively. This fact indicates that the diffusion rate should be the limiting process during the dissolution of ABPBI in the selected complex solvents. The role of the kinetic factor can be seen more clearly when estimating the velocity of “the subsequent diffusion” front, i.e., moving the already dissolved polymer solution from the fiber into the solvent medium. According to data given in Figure A4, a clear boundary of the front was determined, and the dependence of its propagation is shown in Figure 5b.

At the initial moment, the speed of the diffusion front is constant for all solvents and is equal to 3.1 ± 0.3 µm∙s^−1^. Approximately 40 s later, a bifurcation point appears (it is shown by the arrow in Figure 5b), where the front propagation velocity decreases, but in a different way for different dissolving systems. Quantitative features of this process are reflected in Table 2.

The moment of diffusion speed change coincides with the completion of the fiber swelling (stage II in Figure 5a). A significant slowdown of the diffusion front speed can be explained by a combination of two factors. The first one is an increase in the viscosity of the solvent due to a decrease in the solvent composition of low-viscosity methanol, which causes the depletion of the proportion of potassium methoxide, which solvates the polymer in the dissolution zone. The second factor is the heterogeneity of the ABPBI structure, which is a partially crystalline polymer. In this case, the initial dissolution of the amorphous phase, followed by the dissolution of the crystalline one takes place.

Considering the first factor, one should keep in mind that in the boundary of the diffusion front, polymer molecules are immersed in a dilute solution with viscosity depending on composition.

Hence, it is fair to assume that in the case when the rate of swelling and dissolution of the polymer significantly exceeds the diffusion rate of the forming solution into the solvent in the dissolution zone, then there should be a direct correlation between the diffusion front rate and the solvent viscosity.

Therefore, to test the first hypothesis, the solvents’ viscosities were measured and compared with the diffusion front rate (complete solvent flow curves are presented in Appendix A Figure A5).

The study of three dissolving systems suggests that a direct correlation exists between the solvent viscosity and the diffusion front rate, which corresponds to previous knowledge (Figure 6) [37].

To test the second hypothesis, the X-ray diffraction patterns of the fibers were taken before and after the stage of washing out the amorphous part (exposure to the solvent for 40 s).

One can see the existence of a characteristic peak at 26 degrees in the case of the meridian direction in the transmission mode, an increase in the peak intensity in the same 2Θ range in the reflection mode, as well as a slight narrowing of reflections in the equator direction after the selective dissolution of the amorphous phase in the fiber.

The selective dissolution of the more amorphous phase of the fiber is well observed considering the fiber through the linear crossed polaroid (Figure 7b I and II).

On the fiber washed by the solvent after stage II, fibrillarity and the presence of areas with different degrees of orientation are observed, as indicated by non-uniform light transmission. It should be noted that such morphology is not exclusively a feature of the oriented fiber; a similar pattern is also observed in the case of polymer powder, from which solutions are prepared (Figure 7b III).

Thus, the dissolution kinetics is largely determined by the dissolution rate of the more oriented phase presented in the polymer fiber. This occurs most rapidly in the methanol-enriched mixture, which is probably associated with a higher concentration of methylate in the dissolution zone.

### 3.4. Rheology of Concentrated Solutions

The rheological behavior of solutions with the same ratios of DMSO:MeOH as before at a constant KOH content was studied in detail for 3, 7, and 9% solutions at 20 °C. The corresponding data are shown in Figure 8.

The behavior of 3 and 7% solutions in different solvents is qualitatively similar to each other. On the flow curves of solutions, a plateau of the highest Newtonian viscosity is observed, that narrows as the polymer concentration increases (Figure 8a). An increase in the methanol concentration relative to DMSO leads to a slight decrease in viscosity, elastic, and loss moduli. As the methanol concentration is decreased, the crossover point slightly shifts to the region of lower frequencies (Figure 8b).

An increase in the polymer concentration to 9% leads to the appearance of qualitative differences in the behavior of solutions in all solvents investigated. Solution A9 is a highly structured system with a yield point of ~3500 Pa and close values of the complex dynamic modulus components. On the frequency dependences of the moduli, the crossover point is clearly expressed already at a frequency of 4 rad∙s^−1^. Increasing the methanol to DMSO up to 60% (sample A10) leads to a change in the rheological behavior of the solution. A pronounced area of the highest Newtonian viscosity up to 10 s^−1^ appears in the flow curve. On the frequency dependences of moduli, the crossover point shifts to the region of high frequencies (up to 150 rad∙s^−1^) with a significant (up to ~10 times) decrease in moduli values (Figure 8b).

More clearly, the difference in the behavior of the solutions is seen when comparing the tangents of the frequency dependences of the modules in logarithmic coordinates in the terminal zone (Figure 9).

For 3% and 7% solutions, the slopes almost do not change depending on the methanol to DMSO ratio, decreasing as the polymer concentration increases and are equal for 3% solution to 1.75 (*G′*) and 1.00 (*G″*), and 7% solution to 1.35 (*G′*) and 0.9 (*G″*). With an increase in the polymer concentration to 9%, the tangents on the frequency dependences for both moduli decrease, while the system with 10% methanol is highly structured, with the tangents of the frequency dependences of the elastic moduli and losses equal to 0.5 and 0.4, respectively. For a solution with a content of methanol in the solvent of 60%, the slope angles are more than twice as high—1.1 (*G′*) and 0.75 (*G″*).

The observed rheological behavior of solutions can be explained by the mechanism of polymer dissolution. In the case of a 7% polymer content, the amount of methylate with potassium dimsyl is sufficient for the limiting solvation of the polymer, while with an increase in concentration to 9%, part of the functional groups of the polymer remains incompletely solvated and is capable of interacting with each other, forming a dense network similar to that observed in a poor solvent [38].

Thus, the role of methanol is mainly manifested in concentrated solutions, in which an increase in the methanol fraction to 60% contributes to the formation of less structured solutions, from which films and fibers can be obtained easily.

#### Temperature Effect

The upper temperature range of the studied solutions is limited by the methanol boiling point (64 °C). Therefore, the measurement of rheological characteristics was carried out at temperatures not exceeding 50 °C. The flow curves and frequency dependences of the moduli at 20, 35, and 50 °C for a chosen 7% solution are presented in Figure 10.

With the temperature increasing, the character of the flow almost does not change. When temperature arising from 20 to 50 °C, the viscosity in the Newtonian region decreases three times, and the slopes of the elastic and loss moduli in the frequency dependences in the terminal zone remain unchanged and equal to 1.4 and 0.9, respectively, which indicates the same type of relaxation properties of the system at different temperatures. Temperature variation does not allow significant control over the rheological properties of such systems.

The viscous flow activation energy values were calculated using the Arrhenius equation:$$\eta =A\;exp\left(\frac{E_{act}}{RT}\right),$$
where *E_act_*, *R*, *T* and *A* are the flow activation energy (J/mol), the universal gas constant (8.314 J/mol·K), the absolute temperature (K) and the viscosity constant (Pa-s), respectively. In accordance with this, the values were calculated from viscosity values at 4 different temperatures for all solutions as:$$\tan\alpha =\frac{d\left(\ln\eta \right)}{d\left(T^{-1}\right)}=\frac{E_{act}}{R}.$$

Figure 11 shows the *E_act_* values as a function of the methanol concentration in the solution for different concentrations of ABPBI.

As can be seen from the figure, methanol plays a significant role in the complex organic solvent, which becomes more significant as the polymer concentration increases. Thus, an increase in the proportion of methanol relative to DMSO from 10 to 60% has almost no effect on *E_act_* for 3% solutions but reduces *E_act_* for 7% solutions by 1.5 times and for 9% solutions by 2.3 times.

A probable explanation for this phenomenon is the role of methanol as an intermediate solvent compatible with DMSO and necessary for the dissolution of alkali, which is practically insoluble in pure DMSO. Increasing the polymer concentration requires more alkali, and a larger content of methanol to dissolve it. In the case of its deficiency, the structuring of the system increases.

## 4. Conclusions

It has been shown the methanol in the complex solvent containing MeOH-DMSO-KOH is the necessary component that ensures the formation of a sufficient amount of potassium methoxide which solvates the polymer. On the other hand, its excess can lead to a decrease in the basicity of the solvent medium, turning it into a polymer non-solvent. This is well observed by accelerating the polymer dissolution, and by reducing the structurization of solutions in the best complex solvent containing an optimal amount of methanol about 60 wt.% of the DMSO, that opens a possibility making high concentrated solutions up to 9%. The possibility of obtaining more concentrated ABPBI solutions (above 10%) in the developed three-component solvent is limited by the need to use a higher concentration of KOH poorly dissolved in DMSO, and a corresponding decrease in the basicity of the medium due to the water appearance in the reactions of superbases formation, i.e., potassium methoxide and (to a lesser extent) potassium dimsyl.

The study of the polymer dissolution kinetics, accompanied by the structure investigation, showed the presence of strong crystalline and weak amorphous phases in the ABPBI fiber. The key stage is the dissolution of the most ordered phase, which occurs most rapidly in a solvent with a 60% methanol content.

It is shown that the rheological properties of ABPBI solutions change insufficiently in the temperature range limited by the low boiling point of methanol.

The found solvent composition will allow us to further develop a method for obtaining high-strength heat-resistant ABPBI fibers and films from a high-molecular polymer.

## Figures and Tables

**Figure 1 polymers-14-04648-f001:**
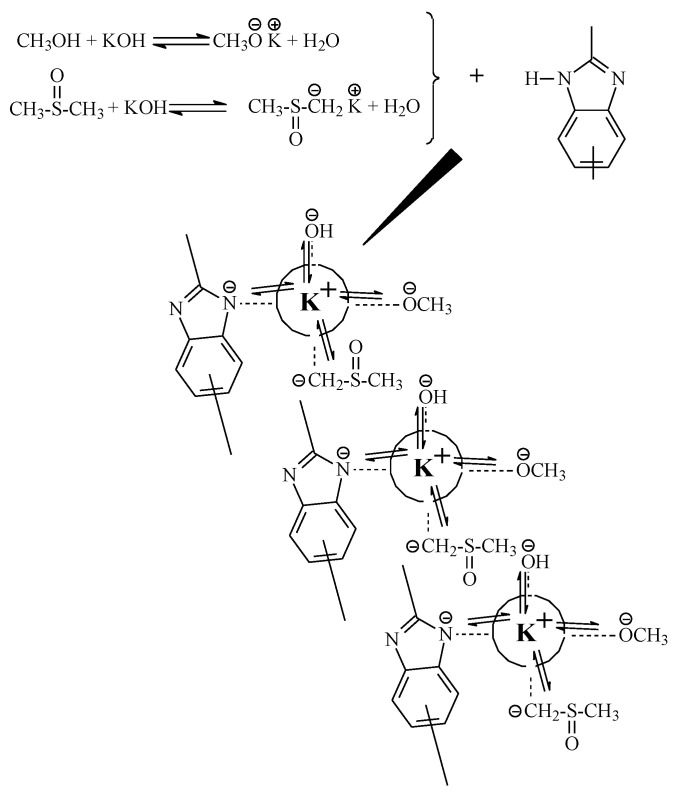
ABPBI dissolution scheme in a complex solvent.

**Figure 2 polymers-14-04648-f002:**
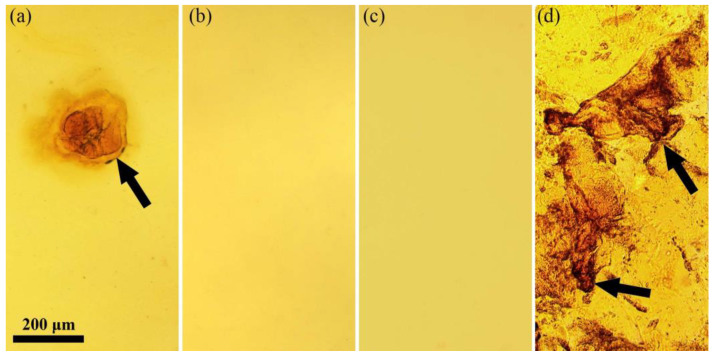
Morphology of 7% ABPBI solution at DMSO:MeOH ratios equal to 95:5 (**a**); 94:6 (**b**); 30:70 (**c**); and 26:74 (**d**). The arrow shows the gel-like particles of the swollen polymer.

**Figure 3 polymers-14-04648-f003:**
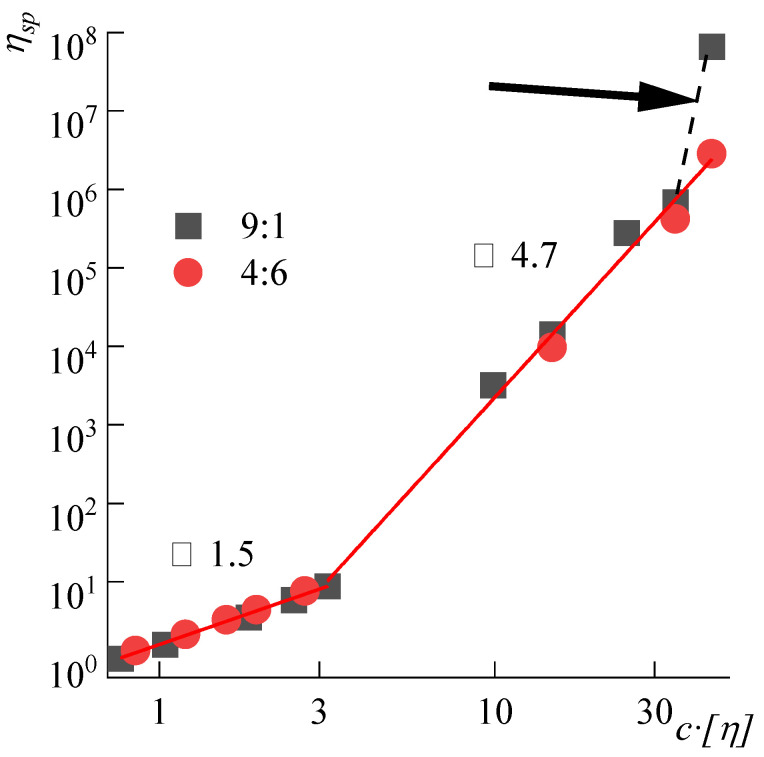
Dependences of the reduced viscosity on the *c[η]* parameter for ABPBI solutions in the DMSO:MeOH:KOH complex solvent with DMSO:MeOH ratios are 9:1 and 4:6, KOH is 100 wt% of the polymer at 25 °C. The arrow shows the deviation from the linear dependence caused by the solution gelation.

**Figure 4 polymers-14-04648-f004:**
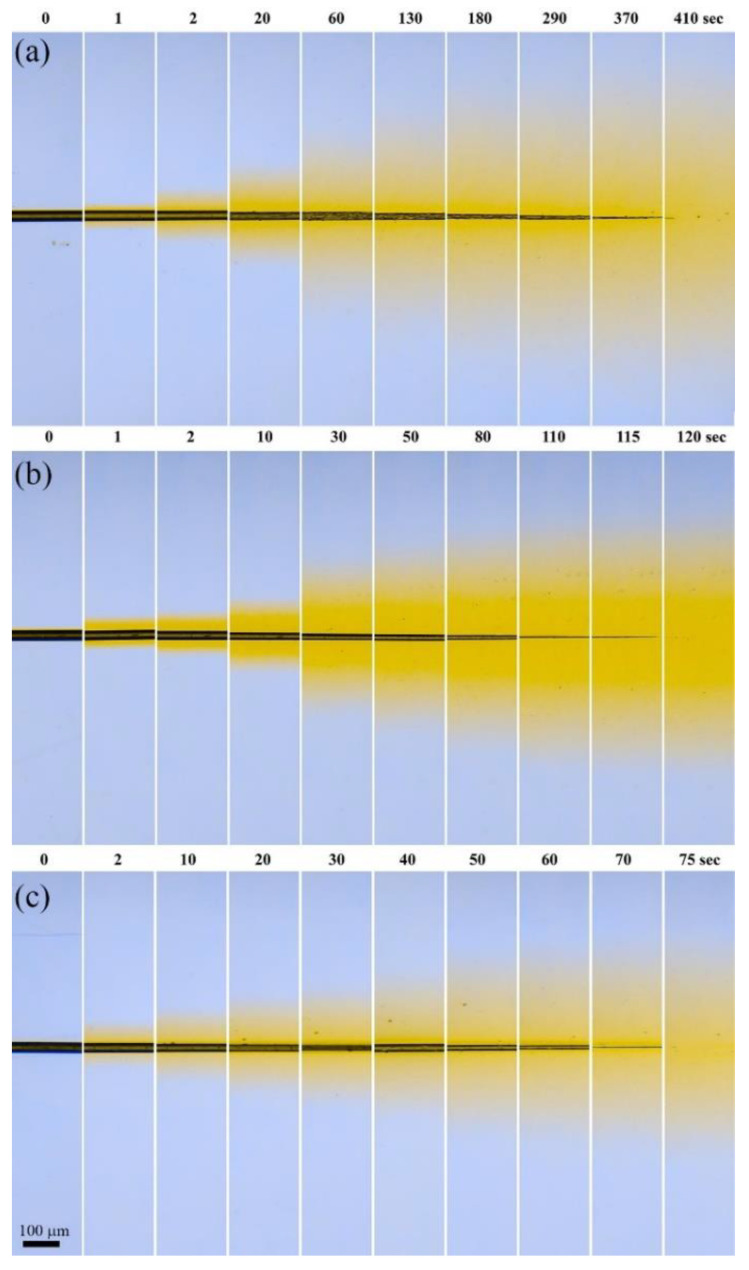
ABPBI fiber dissolution stages in a DMSO:MeOH:KOH complex solvent at DMSO:MeOH ratio: 9:1 (**a**); 7:3 (**b**); 4:6 (**c**).

**Figure 5 polymers-14-04648-f005:**
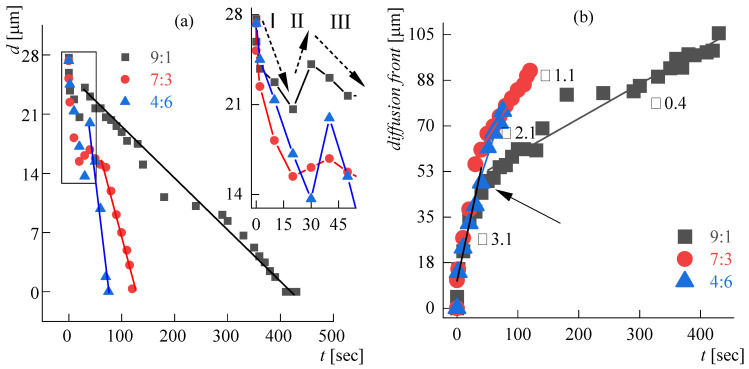
(**a**) Fiber dissolving kinetics in solvents with different ratios of DMSO:MeOH. Stage I corresponds to the initial dissolution of the surface layers, stage II to swelling of the bulk part of the fiber, and stage III to the monotonous subsequent dissolution of the fiber; (**b**) diffusion front propagation of a polymer solution in a solvent medium. DMSO:MeOH ratios are 4:6, 7:3, 9:1; KOH is 100 wt% of the polymer.

**Figure 6 polymers-14-04648-f006:**
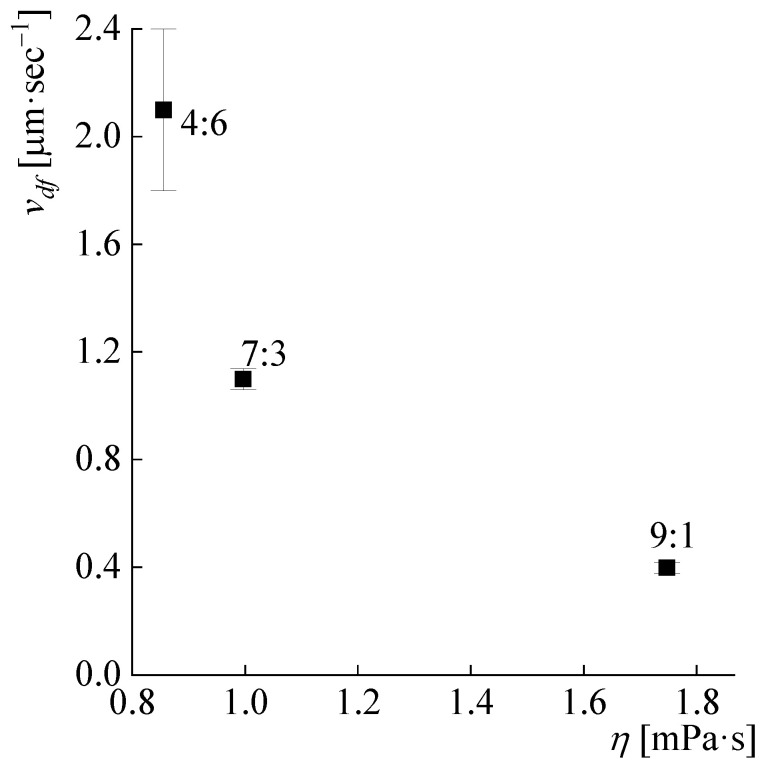
Dependence of the forming solution diffusion rate into the solvent on its viscosity. DMSO:MeOH ratios are 4:6, 7:3, 9:1; KOH is 100 wt% of the polymer.

**Figure 7 polymers-14-04648-f007:**
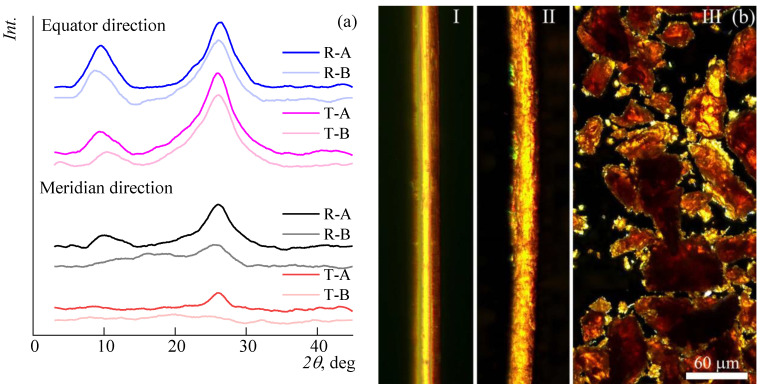
(**a**) X-ray diffraction patterns of ABPBI fibers in equator and meridian directions before (B) and after (A) partial dissolution at reflection (R) and transmission (T) modes; (**b**) ABPBI fiber before (I) and after (II) partial dissolution, and the powder (III) images in linear crossed polaroids.

**Figure 8 polymers-14-04648-f008:**
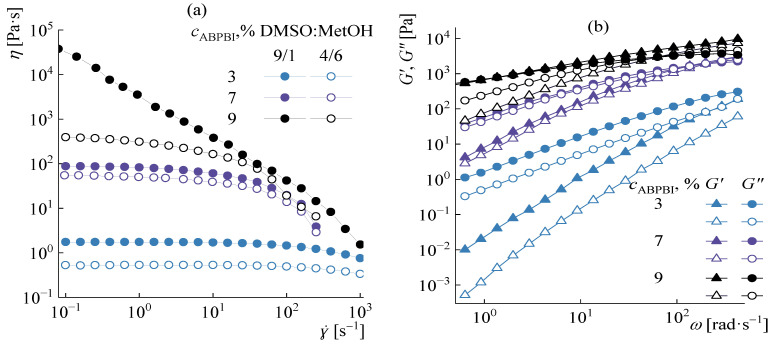
Dependences of viscosity on shear rate (**a**) and elastic and losses moduli on frequency (**b**) for ABPBI solutions. The data for the DMSO:MeOH ratio of 9:1 are shown as filled symbols, and for the ratio of 4:6 they are open.

**Figure 9 polymers-14-04648-f009:**
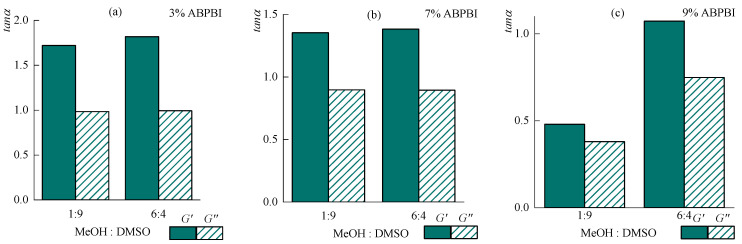
Dependence of the slope tangent of dynamic moduli in the low-frequency range on the MeOH:DMSO ratio, in solutions with a concentration of (**a**) 3, (**b**) 7, and (**c**) 9%.

**Figure 10 polymers-14-04648-f010:**
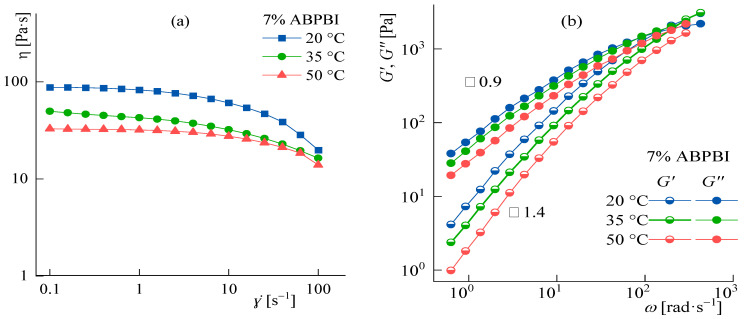
Flow curves (**a**); and frequency dependences of the elastic and loss dynamic moduli (**b**) for 7% solution in a solvent with DMSO:MeOH ratio of 9:1. The slope values in the terminal zone are indicated in the graphs.

**Figure 11 polymers-14-04648-f011:**
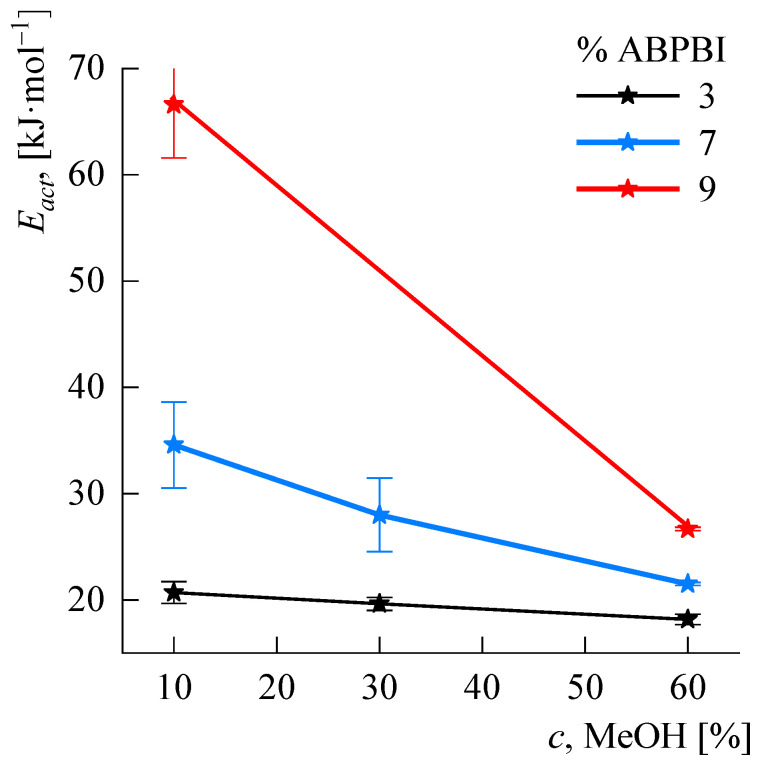
Dependences of the flow activation energy on the content of methanol in the solvent for solutions of different concentration.

**Table 1 polymers-14-04648-t001:** ABPBI solutions composition (wt.%).

Solution Abbreviation	A1	A2	A3	A4	A5	A6	A7	A8	A9	A10
*c* ABPBI, %	3	3	3	3	7	7	7	7	9	9
DMSO:MeOH	9:1	7:3	4:6	3:7	9:1	7:3	4:6	3:7	9:1	4:6
*c* KOH, %	1.5	1.5	1.5	1.5	3.5	3.5	3.5	3.5	4.5	4.5

**Table 2 polymers-14-04648-t002:** Fiber dissolution rates in various solvents.

DMSO:MeOH Ratio	Complete Dissolution Time, s	Diffusion Front Rate, µm∙s^−1^		Fiber Thinning Rate, µm∙s^−1^	
		Before the Bifurcation Point (I)	After the Bifurcation Point (III)	Before the Fiber Swelling (I)	After the Fiber Swelling (III)
9:1	410	3.1 ± 0.3	0.4 ± 0.02	0.29 ± 0.08	0.06 ± 0.001
7:3	120		1.1 ± 0.04	0.55 ± 0.11	0.18 ± 0.02
4:6	75		2.1 ± 0.3	0.44 ± 0.04	0.61 ± 0.04

## Data Availability

The data presented in this study are available on request from the corresponding author.

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
