# Peer review of "High Molecular Weight AB-Polybenzimidazole and Its Solutions in a Complex Organic Solvent: Dissolution Kinetics and Rheology"

_polymers, 2022, doi:10.3390/polym14214648_

Round 1
Reviewer 1 Report
Ivan Y. Skvortsov and co-authors present a detailed study on the dissolution kinetics and rheological characterization of high molecular weight AB-Polybenzimidazole (ABPBI) solutions. The solvent is based in potassium hydroxide (KOH), dimethyl sulfoxide (DMSO) and methanol. The results are clear and they have been well discussed. Indeed, this comprehensive investigation is important for fibbers spinning based in high molecular (ABPBI). In this context, I do think that this manuscript should be published in Polymers. However, before publication, there are small points that I think should be taken into consideration:
1- The authors use “KOH” as solvent but this is just the abbreviation. I don't see anywhere in the manuscript the name of this abbreviation.
2- The same happens with the “DABA”, “DABA MP” and / or “PPA” abbreviations. As far as I could see, I didn’t find what does these abbreviations mean.
3- Page 2, line 83: ”Mw”. The “M” should be in italic.
4- Page 6, line 267: the authors say they cannot use “interferometry method”, however it is not clear, which method was then used by the authors. Please clarify it better.
5- Figure 11: It was not understandable how the authors obtained the activation energies. Please clarify it better.
Author Response
Reviewer 1
Ivan Y. Skvortsov and co-authors present a detailed study on the dissolution kinetics and rheological characterization of high molecular weight AB-Polybenzimidazole (ABPBI) solutions. The solvent is based in potassium hydroxide (KOH), dimethyl sulfoxide (DMSO) and methanol. The results are clear and they have been well discussed. Indeed, this comprehensive investigation is important for fibbers spinning based in high molecular (ABPBI). In this context, I do think that this manuscript should be published in Polymers. However, before publication, there are small points that I think should be taken into consideration:
Thank you very much for taking time to review our manuscript. We appreciate you for detailed consideration of our manuscript. Many constructive suggestions which would help us to improve the quality of it are given. All the changes are highlighted in the revised manuscript and the point-by-point responses to your comments are done.
1- The authors use “KOH” as solvent but this is just the abbreviation. I don't see anywhere in the manuscript the name of this abbreviation.
KOH is a chemical formula of potassium hydroxide; the name is added in the manuscript.
2- The same happens with the “DABA”, “DABA MP” and / or “PPA” abbreviations. As far as I could see, I didn’t find what does these abbreviations mean.
These abbreviations are disclosed.
3- Page 2, line 83: ”Mw”. The “M” should be in italic.
Mw printing is corrected
4- Page 6, line 267: the authors say they cannot use “interferometry method”, however it is not clear, which method was then used by the authors. Please clarify it better.
The interferometry method (described in detail in [31, 32]) is an optical method that consider the interference patterns of two transparent components and interdiffusion zone between them. Evolution of this zone in time allows us to prove the compatibility or solubility of components, e.g. solubility of a polymer in a solvent, and to calculate the diffusion rates of each component. In many cases the estimation of limiting concentrations of interacting components allows plotting the phase diagram of binary system. Due to the fact, that the dried ABPBI films are completely opaque, this method turned out to be inapplicable and we used another method based on determining the kinetics of fiber thinning in a narrow gap. Moreover, for the dark and contrast polymers immersed to a solvent it is possible not only to estimate the dissolution rate, but also to anticipate the difference in the diffusion kinetics of the chosen polymer in different solvents.
The sentence is rewritten now.
5- Figure 11: It was not understandable how the authors obtained the activation energies. Please clarify it better.
The calculation method is added in manuscript text.

Reviewer 2 Report
1. What is the role of the KOH in the process, and could it be replaced with a either NaOH, or an organiz base such as DBU, pyridine etc? In other words, what are the requirements for the base? Some discussion should be added as a guideline.
2. Zeta potential measurements should be rovided for the different systems. How do the developed system alters the zeta potential of the obtained polymer materials?
3. The description of the spinning method, the process are parameters and the equipment setup are not disclosed properly. A separate section under the Experimental part should be dedicated to the spinning for better understanding and reproducibility purposes.
4. The molecular weight of the used polymer should be measured and disclosed in the manuscript. It is a crucial parameter and all the results are affected by the molecular weight of the polymer, therefore it must be measured and reported together with the rest of the results.
5. SEM images of the fibers, as well as the fiber diameter distribution should be reported for the obtained fibers.
6. Elemental microanalysis and EDX should be performed to show that there is no remaining potassium in the fibers left.
7. Commercial polybenzimidazole solutions consists of polar aprotic solvents such as DMAc and salts such as LiCl to aid solubilization and stabilization of the dope solutions. These systems should be compared with the proposed one. What are the pros and cons? Is there any added benefit of the KOH/MeOH/DMSO system?
8. Polybenzimidazole type solutions has multiple uses for spinning and casting, and a list of broad examples should be provided (10.1039/D0GC04077K; 10.1016/j.memsci.2022.120383; 10.1021/acsapm.1c01926; 10.1016/j.desal.2022.115777).
9. Some tables and figures show data with errors or error bars (e.g. Table 2), however, it is unclear how these errors were derived. The authors should explain in the corresponding figure captions how many independently prepared samples were analyzed. Missing errors for Figures 3, 5, 9, 11 should be added. It is important to demonstrate reproducibility of the polymer synthesis and viscosity performance.
10. The conclusion section is too long and vague. A good amount of results are presented, and compared but only the main research findings should make it into the conclusion section. The main research findings should be briefly summarized in quantitative statements.
11. Unambiguous legends should be added for the figures. For instance, it is not mentioned in any legends what the 9:1, 4:6, 1.5 and 4.7 values are in Figures 3 and 5 etc.
Author Response
Reviewer 2
Thank you very much for taking time to review our manuscript. Many constructive suggestions of reviewer kept in attention. All the changes are highlighted in the revised manuscript and the point-by-point responses are listed below. We hope the revised manuscript and the response have addressed directly the reviewer’s comments and suggestions.
- What is the role of the KOH in the process, and could it be replaced with a either NaOH, or an organiz base such as DBU, pyridine etc? In other words, what are the requirements for the base? Some discussion should be added as a guideline.
The dissolution of ABPBI requires the creation of a superbasic medium. To do this, it is necessary to minimize the number of protons (H+) in the solvent and use a strong base with the highest ionic radius of the cation. Therefore, K is more preferable in comparison with Na (we carried out experiments in NaOH, but the solubility of high molecular weight ABPBI in it was much worse). Perhaps RbOH or CsOH would be even more preferable, but it is significantly more expensive, and we wanted to find a formulation that was suitable not only for laboratory conditions. We did not consider the DBU and pyridine due to the toxicity of these reagents.
- Zeta potential measurements should be rovided for the different systems. How do the developed system alters the zeta potential of the obtained polymer materials?
Traditionally, z-potential relates to sliding plane of the double electric layer, i.e. to colloid systems. We have the homogeneous solutions. Sometimes, the macromolecular coil is considered as a particle, especially in the charged solvent. Following your suggestion, we tried to perform the short set of measurements. To estimate the z-potential value, it is necessary to know the parameters of the medium – the viscosity and dielectric constant. In a flat capacitor with an intrinsic capacitance of 50 pF and losses (D) <0.001, the capacitance of two solvents (DMSO:methanol, equal to 9:1 and 6:4) was measured over a wide frequency range.
It is seen that solvents have very large values ​​of the loss tangent, which increases with increasing frequency. The values ​​of the dielectric constant in the low frequency region are extremely high - about 10^4, which is probably due to the edge effects and requires the development of a separate measurement technique, which is unavailable in our laboratory.
Therefore, we used the approximate values ​​of the permittivity required to calculate the zeta potential for DMSO-methanol mixtures 9:1 and 6:4 is equal to 46 and 45 (according to https://doi.org/10.1016/j.fluid.2013.12.014), making the assumption that the presence of an alkali changes the dielectric constant in the same way for both solvents. This allows us to estimate roughly the values ​​of the zeta potentials for ABPBI solutions in 9:1 and 4:6 solvents. The values ​​were -26 and -18 mV, respectively. This indicates a decrease in the stability of solutions with increase of methanol content, that can be explained by a decrease in the basicity of the superbase. This is only a qualitative result, but sufficient for the first approximation.
- The description of the spinning method, the process are parameters and the equipment setup are not disclosed properly. A separate section under the Experimental part should be dedicated to the spinning for better understanding and reproducibility purposes.
The main aim of this work is the choice of the optimal solvent for ABPBI, estimation of mechanism of dissolution and the properties of prepared solutions. The fibers were spun from solutions in described set of components, but without detailed analysis of the spinning process. That is why these data are not yet worthy of publication. In our experiment, it is only important that all the fibers were of the same thickness and obtained under the same conditions in order to compare effect of the solvent. To avoid confusion and misunderstanding, the section with such a brief description of spinning procedure has been changed.
- The molecular weight of the used polymer should be measured and disclosed in the manuscript. It is a crucial parameter and all the results are affected by the molecular weight of the polymer, therefore it must be measured and reported together with the rest of the results.
One of the direct and universal parameters characterizing the size of a polymer coil in a solvent is the intrinsic viscosity. It is directly related to the molecular weight in the same solvent. As the most accurate and well reproducible parameter, namely the intrinsic viscosity was chosen as parameter characterized dimensions of macromolecules.
As for the molecular weight values, in [2] the molecular weight values ​​of the ABPBI samples were measured by the dynamic light scattering method. Based on these data, we can say that polymer used in this work has an Mw of the order of 100,000 g/mol.
- SEM images of the fibers, as well as the fiber diameter distribution should be reported for the obtained fibers.
The purpose of this work was to study the solubility of the polymer, we deliberately do not present data on the fibers spinning, their structure and properties, because we plan to publish these data in future. Here we would like to concentrate attention on the choice of the optimal solvent, which will make it possible to obtain highly concentrated spinning solutions for subsequent defect-free spinning of fibers.
- Elemental microanalysis and EDX should be performed to show that there is no remaining potassium in the fibers left.
We agreed, this is a very important parameter, but for the spun fibers. This paper did not touch in details spinning procedure, properties and compositions of fibers.
- Commercial polybenzimidazole solutions consists of polar aprotic solvents such as DMAc and salts such as LiCl to aid solubilization and stabilization of the dope solutions. These systems should be compared with the proposed one. What are the pros and cons? Is there any added benefit of the KOH/MeOH/DMSO system?
High molecular weight ABPBI are not dissolved in aprotic solvents with LiCl or do not allow to obtain a high concentration of polymer in solution, suitable for fiber production. In addition, even for traditional polybenzimidazole solutions, commercially significant solvents are being sought with the exception of lithium salts, the cost of which has increased significantly in recent years.
- Polybenzimidazole type solutions has multiple uses for spinning and casting, and a list of broad examples should be provided (10.1039/D0GC04077K; 10.1016/j.memsci.2022.120383; 10.1021/acsapm.1c01926; 10.1016/j.desal.2022.115777).
PBI is a fairly broad class of polymers. To provide all works devoted to obtaining fibers is impossible even in a review on this topic. So, we tried to cite the most significant publications, in our opinion, dedicated specifically to ABPBI. Suggested sources have been added to the References of the article.
- Some tables and figures show data with errors or error bars (e.g. Table 2), however, it is unclear how these errors were derived. The authors should explain in the corresponding figure captions how many independently prepared samples were analyzed. Missing errors for Figures 3, 5, 9, 11 should be added. It is important to demonstrate reproducibility of the polymer synthesis and viscosity performance.
The deviations in Table 2 were calculated as the average deviation when fitting the experimental data with a linear function (root of variance). These errors were calculated using the standard tools of the Origin Lab program. The fiber thinning data were studied for a series of 3 samples, which showed fairly good reproducibility.
Viscosity measurement errors do not exceed 1% of the measured value (in a case of correct measurement conditions). Therefore, the error bars would be completely invisible in this diagram (8 orders of magnitude along the viscosity axis!)
Figure 5 shows the experimental points, which, in our opinion, indicate quite well the trends in fiber thinning. Adding error bars to this plot will significantly reduce its readability. Moreover, the estimated data errors are shown in Table 2.
The data presented on Fig.9 are based on well-reproducible data of frequency dependences of elastic and loss moduli. Adding error bars could improve the reliability, but will significantly complicate the perception of the Figure.
The error bars were added in Fig. 11.
- The conclusion section is too long and vague. A good amount of results are presented, and compared but only the main research findings should make it into the conclusion section. The main research findings should be briefly summarized in quantitative statements.
The conclusion section was rewritten.
- Unambiguous legends should be added for the figures. For instance, it is not mentioned in any legends what the 9:1, 4:6, 1.5 and 4.7 values are in Figures 3 and 5 etc.
The correspondence legends were added.
Reviewer 3 Report
Dear authors,
This manuscript presents interesting results. It is recommended to review the following comments:
- Please check grammar of lines 14, 24, 222, 240
- The hypothesis or novelty of the work is not clear
- Why did you use different geometries for viscosity measurement?
- How many repetitions were made for the rheological measurements? Taking into account that the results must be representative of the system under study
- The experimental conditions of XRD are not found in the methodology section
- In figure 11, it seems that there is a data missing for the red curve
- What is the perspective of the research work?
Author Response
Thank you very much for taking time to review our manuscript entitled. We would like to appreciate you for your work on our manuscript. Many constructive suggestions which would help us to improve the quality of the paper are given. All the changes are highlighted in the revised manuscript and the point-by-point responses to the comments of you are below.
Dear authors,
This manuscript presents interesting results. It is recommended to review the following comments:
1) - Please check grammar of lines 14, 24, 222, 240
The grammar checked and corrected.
2) - The hypothesis or novelty of the work is not clear
In this work, the conditions and mechanism of dissolution of high molecular weight ABPBI in a complex organic solvent were studied for the first time, and the effect of each of the solvent components on the dissolution kinetics and rheological properties of solutions was determined.
3) - Why did you use different geometries for viscosity measurement?
We investigated the viscosities of solutions in the viscosity range of 8 decimal orders. This is only possible when choosing a series of operating units to be able to obtain sufficient sensitivity (especially for oscillatory measurements of elastic and loss moduli) of low-concentration solutions. The reliable work with highly concentrated solutions requires using small geometry to be in limits of the measurable torque.
Of course, the data obtained on different geometries within their reliable measurement ranges are perfectly reproducible. The usual measurement error of the same sample is less than 1% of the value and is not noticeable in logarithmic coordinates.
4) - How many repetitions were made for the rheological measurements? Taking into account that the results must be representative of the system under study
For rheological studies, from 3 to 15 measurements were performed for each of the solutions and most solutions were prepared 2-5 times from the same polymer. We have been collecting statistics for a long time to obtain adequate and reproducible data to exclude the influence of air moisture, methanol volatility and the conversion of KOH to K2CO3 in air both in the process of preparing solutions and in the process of rheological measurements.
5) - The experimental conditions of XRD are not found in the methodology section
The experimental conditions were added.
6) - In figure 11, it seems that there is a data missing for the red curve
Unfortunately, we did not make the experiments with 9% ABPBI solution in the solvent containing 70% DMSO and 30% MeOH due to it quite poor solubility.
7) - What is the perspective of the research work?
The found solvent composition will allow us to further develop a method for obtaining high-strength heat-resistant ABPBI fibers and films from a high-molecular polymer.

Round 2
Reviewer 2 Report
The manuscript can be published.